# ECOLOGICAL REINFORCEMENT LEARNING

## ABSTRACT

Reinforcement learning algorithms have been shown to effectively learn tasks in a variety of static, deterministic, and simplistic environments, but their application to environments which are characteristic of dynamic lifelong settings encountered in the real world has been limited. Understanding the impact of specific environmental properties on the learning dynamics of reinforcement learning algorithms is important as we want to align the environments in which we develop our algorithms with the real world, and this is strongly coupled with the type of intelligence which can be learned. In this work, we study what we refer to as ecological reinforcement learning: the interaction between properties of the environment and the reinforcement learning agent. To this end, we introduce environments with characteristics that we argue better reflect natural environments: non-episodic learning, uninformative "fundamental drive" reward signals, and natural dynamics that cause the environment to change even when the agent fails to take intelligent actions. We show these factors can have a profound effect on the learning progress of reinforcement learning algorithms. Surprisingly, we find that these seemingly more challenging learning conditions can often make reinforcement learning agents learn more effectively. Through this study, we hope to shift the focus of the community towards learning in realistic, natural environments with dynamic elements.

## 1 INTRODUCTION

A central goal in current AI research, especially in reinforcement learning (RL), is to develop algorithms that are general, in the sense that the same method can be used to train an effective model for a wide variety of tasks, problems, and domains. In RL, this means designing algorithms that can solve any Markov decision process (MDP). However, natural intelligence – e.g., humans and animals – exists in the context of a natural environment. People and animals cannot be understood separately from the environments that they inhabit any more than brains can be understood separately from the bodies they control. In the same way, perhaps a complete understanding of artificial intelligence can also only be obtained in the context of an environment, or at least a set of assumptions on that environment.

There has been comparatively little study in the field of reinforcement learning to understand how properties of the environment impact the learning process for complex RL agents. Many of the environments used in modern reinforcement learning research differ in fundamental ways from the real world. First, standard RL benchmarks, such as the arcade learning environment (ALE) (Bellemare et al., 2013) and Gym (Brockman et al., 2016) are *episodic*, while natural environments are continual and lack a "reset" mechanism, requiring an agent to learn through continual interaction. Second, most of these environments include detailed reward functions that not only correspond to overall task success, but also provide intermediate learning signal, thus shaping the learning process. These signals can aid in learning, but they can also bias the learning process. Third, the environments are typically static, in the sense that only the agent's own actions substantively impact the world. In contrast, natural environments are stochastic and dynamic: an agent that does nothing will still experience many different states, due to the behavior of natural processes and other creatures.

In this paper, we aim to study how these properties affect the learning process. At the core of our work is the concept of *ecological* reinforcement learning: the idea that the behavior and learning dynamics of an agent, like that of an animal, must be understood in the context of the environment in which it is situated. We therefore study how particular properties of the environment can facilitate

Figure 1: We study three properties of realistic environments that can have a large effect on the difficulty of reinforcement learning: (left) non-episodic learning, where the agent is not reset automatically to the same initial state distribution, but must handle whatever situation it puts itself in; (middle) environment shaping as an alternative to reward shaping, where the agent has a single, sparse reward, but the environment is varied so as to provide a curriculum (e.g., due to its natural dynamics, or by a cooperative teacher agent); (right) dynamic environments, where the environment changes due to the actions of other agents and natural phenomena, even if the agent does not take a coordinated course of action – such dynamic phenomena can, as we will show, alleviate some of the difficulties in non-episodic learning.

or harm the emergence of complex behaviors. We focus our attention on the three properties outlined above: (1) continual, non-episodic environments where the agent must learn over the course of one "lifetime," (2) environments that lack detailed reward shaping, but instead provides a reward signal based on a simple "fundamental drive," (3) environments that are inherently dynamic, evolving on their own around the agent even if the agent does not take meaningful or useful actions. We study how each of these properties affects the learning process. Although on the surface these properties would seem to make the learning process harder, we observe that in some cases, they can actually make reinforcement learning *easier*.

The degree to which these properties make learning easier is highly dependent on the degree of scaffolding that is provided by an environment. For example, an agent tasked with collecting and making food pellets might struggle to learn if it must first complete a complex sequence of actions. However, if food pellets are initially plentiful, the agent can first learn that food pellets are rewarding, and then gradually learn to make them out of raw ingredients as the initial supply becomes scarce. This provides a natural scaffolding and curriculum without requiring manual reward engineering. More generally, "environment shaping" can be used as a way to craft the agent's curriculum without modifying its reward function. This benefit is counter-balanced by the fact that non-episodic learning is inherently harder – the resets in episodic tasks provide a more stationary learning problem, preventing the agent from getting "stuck" due to a bad initial policy. However, natural environments can also counteract this difficulty: a dynamic environment that gradually changes on its own can provide a sort of "soft" reset that can mitigate the difficulties of reset-free learning, and we observe this empirically in our experiments. We illustrate some of these ideas in Figure 1.

The contribution of this work is an empirical study of how the properties of environments – particularly properties that we believe reflect realistic environments – impact reinforcement learning. We study the effect of (1) continual, non-episodic learning, (2) learning with and without reward shaping, and (3) learning in dynamic environments that evolve on their own. We find that, though each of these properties can make learning harder, they can also be combined in realistic ways to actually make learning *easier*. We also provide an open-source environment for future experiments studying "ecological" reinforcement learning, and we hope that our experimental conclusions will encourage future research that studies how the nature of the environment in which the RL agent is situated can facilitate learning and the emergence of complex skills. This exercise helps us determine which types of algorithmic challenges we should focus our development efforts towards in order to solve *natural* environments that agents might encounter.

## 2 RELATED WORK

Solving general RL problems can be extremely hard in general (Kakade & Langford, 2002). Reward shaping is a common technique to guide learning (Ng et al., 1999; Devlin & Kudenko, 2012; Brys et al., 2015) but is usually hand crafted and must be carefully designed by human experts (Griffith et al., 2013). Shaping the reward may also lead to suboptimal solutions, as it alters the objective of the learning problem. Curriculum learning can be used to first provide the agent with easier tasks, followed by more challenging tasks (Bengio et al., 2009; Graves et al., 2017; Randløv & Alstrøm, 1998; Wang et al., 2019a; Yu et al., 2018; Heess et al., 2017). Curriculum learning can also be viewed in the context of multiple learning agents in an adversarial or cooperative setting (Silver

et al., 2016; Al-Shedivat et al., 2017; Sukhbaatar et al., 2017; Omidshafiei et al., 2018) or where the curriculum is automatically generated (Florensa et al., 2017b;a; Riedmiller et al., 2018; Wang et al., 2019b). The "environment shaping" that we study in our experiments can be viewed as a kind of curriculum learning, and we argue – and show empirically – that this environment shaping approach can in some cases be more effective than more commonly used reward shaping.

Improved exploration methods are a possible solution to solving sparse reward tasks. Prior work has used approximate state-visitation counts (Tang et al., 2016; Bellemare et al., 2013), information gain, or prediction error (Houthooft et al., 2016; Pathak et al., 2017), or model ensemble uncertainty (Osband et al., 2016). A recent work (Ecoffet et al., 2019) maintains a set of novel states and first returns to the novel states before exploring from this frontier. Our work could be combined with an exploration method, however, this work indicates that sparse reward tasks can be solved with an appropriately shaped environment.

Prior work on RL without resets has focused on safe exploration (Moldovan & Abbeel, 2012; Chatzilygeroudis et al., 2018) or learning a policy to reset the environment (Eysenbach et al., 2017; Han et al., 2015). Even-Dar et al. (2005) studies reset free RL in POMDPs and implements a homing strategy which approximately resets the agent. Rather than trying to convert the reset-free problem to one that looks more like a scenario with resets, our experiments study under which conditions reset-free learning can actually be easier, and show that dynamic environments – which we argue better reflect the real world – actually make learning without resets easier.

Learning in non-episodic settings has been studied from the perspective of continual learning Ring (1997), where a number of tasks are learned in sequence. These algorithms typically consider the problem of "catastrophic forgetting" (Mccloskey, 1989; French, 1999), where previously learned tasks are forgotten while learning new tasks. To solve this problem, algorithms use methods such as explicit memorization (Rusu et al., 2016; Schwarz et al., 2018), generative replay (Shin et al., 2017) and explicit weight regularization (Kirkpatrick et al., 2016; Kaplanis et al., 2018). These works assume that resets and task boundaries are available whereas we assume that the agent is unable to reset. There has also been work on building more complex tasks in large diverse worlds with Mujoco (Todorov et al., 2012; Singh et al., 2019; Yu et al., 2019), Malmo (Johnson et al., 2016; Guss et al., 2019), DeepMind Lab (Beattie et al., 2016), and many others, however, again, these environments are studied in the context of episodic-learning.

## 3 PROPERTIES OF NATURAL ENVIRONMENTS

In contrast to most simulated environments that are used for reinforcement learning experiments (Brockman et al., 2016), agents learning in natural environments experience a continual stream of experience, without episode boundaries. The typical reward function engineering that is often employed in reinforcement learning experiments is also generally unavailable in the real world, where agents must rely on their own low-level perception to understand the world. Finally, natural environments change on their own, even when the agent does not follow a coordinated or intelligent course of action. This dynamism can create additional challenges, but can also facilitate learning, mitigating some of the issues due to non-episodic and non-resettable learning settings. In this paper, our aim is to study how these aspects of the environment impact the performance of reinforcement learning agents. We term this approach *ecological* reinforcement learning, in that it deals specifically with the relationship between properties of the environment and the reinforcement learning agent, rather than studying reinforcement learning algorithms in the general case, regardless of the particular properties of the learning environment. We believe that the properties outlined above are broadly reflected in real-world settings, and are often absent in simulated reinforcement learning benchmarks. In this section, we discuss each of these properties, and formulate our hypotheses about how these properties might influence learning.

**Continual non-episodic learning.** In the real world, all learning must at some level be non-episodic: though we may instrument environments to make them *appear* episodic, there is always a single underlying temporal process. In general, this makes the learning problem harder: when the agent is not reset to randomly chosen initial states, mistakes early on in training can put it into undesirable situations, from which it might be harder to recover and – more importantly – harder

to learn. A non-episodic learning process is non-stationary, and the agent can become trapped in difficult regions of the state space.

*Hypothesis 1*: Non-episodic learning is more difficult than episodic learning because the agent must handle a non-stationary learning problem, and can become trapped in difficult states. We will study this hypothesis in our experiments, and show how some of the other properties of natural environments can help alleviate this difficulty.

**Sparse rewards and environment shaping.** While in principle RL algorithms can handle relatively uninformative rewards, in practice reward shaping is often an essential tool for getting RL methods to acquire effective policies. For example, an agent that must learn a policy to collect resources to make an axe (see Figure 2) might make use of a reward function that specifies the distance to the nearest resource, or at least provides a small reward for each resource obtained, as opposed to a reward given only for obtaining the final goal. However, well-shaped rewards are generally not available and difficult to provide in the real world, since they require knowledge of privileged state variables (e.g., positions of objects) or the process by which the task must be completed (e.g., required resources), both of which should in principle be learned automatically by the agent. Furthermore, reward shaping might introduce bias, since the optimal policy for a shaped reward may not in fact be optimal for the original task reward. On the other hand, agents in the real world do not learn in a vacuum: even for humans and animals, it is reasonable to assume a reasonably *cooperative* environment that has been set up so as to facilitate learning. For humans, this kind of "scaffolding" is often provided by other agents (e.g., parents and teachers). But even without other agents, natural environments might provide automatic scaffolding – e.g., an animal might find apples that fell from a tree, and thereby learn that apples are a source of food. Once the fallen apples are exhausted, the animal might use its knowledge of the value of apples to learn to climb the tree to obtain the apples on its own. This kind of "environment shaping" could serve as a tool for guiding the learning process, without the bias or manual engineering inherent in reward shaping.

*Hypothesis 2*: Environment shaping can enable agents to learn even with simple sparse rewards, and can in fact result in more proficient policies if applied correctly, as opposed to reward shaping.

**Dynamic environments.** Standard reinforcement learning benchmark tasks are typically situated in *static* environments (Brockman et al., 2016; Bellemare et al., 2013), in the sense that the environment does not change substantially unless the agent takes a coordinated course of action. On the other hand, real-world settings are typically *dynamic*, in the sense that the environment changes even if the agent does not follow any coordinated course of action: animals will move around, times of day will change, seasons will change, etc. Dynamic environments present their own challenges, but they can also facilitate learning, by automatically exposing the agent to a wide variety of situations.

*Hypothesis 3*: While dynamic environments could make learning more difficult, in fact they can alleviate some of the challenges associated with non-episodic learning, by providing the agent with a variety of learning conditions even in the absence of coordinated and intelligent behavior (as is the case, e.g., early on in training).

## 4 EXPERIMENTAL SETUP

To carry out our ecological RL study, we construct three simulated tasks. The simulator for two of them is built on top of the grid-like environment proposed by Chevalier-Boisvert et al. (2018). We chose a grid-based discrete-action environment over a more complex, high-dimensional one to study the aforementioned properties in isolation, without other confounding factors involving high-dimensional observations and representation learning. The goal is not to simulate a completely visually realistic and life-like system, but to study those properties of the MDP that will be particularly important in natural environments, and have not been addressed in detail in prior work.

The environment is an $N \times N$ grid of tiles, where each tile contains at most one object, but the agent is free to move over any tile and can pick up and carry one object at a time. The agent can use objects in its environment to construct new ones by dropping a carried object onto an existing object. Objects include wood, metal, deer, axe, and food. The agent can combine wood with metal to construct an axe and apply the axe to a deer to produce food. The agent consumes resources such as food by picking them up. There are movement actions associated with each of the cardinal

directions, as well as to pick up or drop an object. The environment is partially observed, and the agent receives a local egocentric view around it, represented by the shaded region in Figure 2, which is a $5 \times 5 \times C$ grid, where $C$ is the number of object types, and each grid position contains a one-hot vector representation of the object type. We use the following two tasks in our evaluation, which are illustrated in Figure 2:

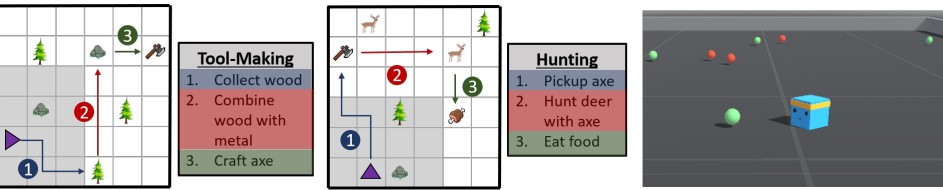

Figure 2: Tasks in our partially observed stochastic environment for crafting an axe (left) and hunting a deer (middle). The agent (purple triangle) receives a local observation (shaded gray square) and must interact with objects in the correct sequences to complete the task. In the food collection task in Unity (right), the agent's goal is to collect green food and avoid red poison.

*Tool-making:* The agent must combine wood and metal to craft an axe, and then pick up the axe. Wood and metal appear at random locations initially, and continue to spawn with some probability as time progresses in the dynamic setting.

*Hunting:* The agent uses an axe to hunt a deer, which produces food that it can pick up to eat. Axes and deer appear at random locations. The deer can move around in the environment in the dynamic setting, and can appear at different distances from the agent.

In order to study our hypotheses in varied settings, we additionally investigate these environmental properties, enumerated at the end of this section, in the context of the Food Collector environment available as part of the Unity ML-Agents toolkit provided by Juliani et al. (2018). The environment features a continuous state space with raycast partial observations representing the directional view of the agent. The action space is 27-dimensional, with separate action streams corresponding to forward, lateral, and rotational movement. The task is taken as-is, wherein the agent must maximize the number of healthy food items eaten (represented by green spheres) while avoiding consuming poisonous food (red sphere). We modify the environment dynamics and initial conditions within this task setting to investigate our hypotheses.

To study the hypotheses discussed in the previous section, we vary a number of properties of these environments, and examine their effect on the learning process:

**Non-episodic learning.** The non-episodic version of each task does not allow the agent to reset to the initial state, and instead requires it to learn the task effectively over one very long episode, as illustrated in the figure on the right. For example, in the Unity Food Collector task, the agent must learn to continually collect as many healthy food items as possible while avoiding the poisonous food across its lifetime. We will compare this against the episodic case, where the agent is reset to an

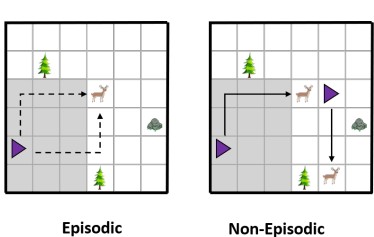

initial state distribution after completing the task or after a fixed time horizon. This will allow us to study hypothesis *H1*.

**Reward shaping.** Reward shaping is often used to assist the agent in finding a good policy by providing more signal to the agent. The environment allows us to experiment with different reward functions, such as a shaped distance-based reward function that rewards the agent for how close it is to the nearest resource it needs, and a larger bonus for interacting with the right object. A sparse reward function will be based on a simple "fundamental drive:" whether or not the agent has just acquired the axe (in the tool-making task), deer (hunting), or food (food collector).

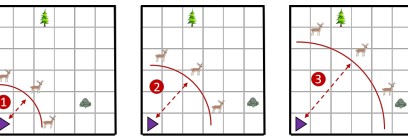

**Environment shaping.** Instead of shaping rewards, which can be unrealistic and can bias the learning process, we can instead change the types and distribution of states the agent observes while

learning. In our tasks, we can scaffold the learning process by controlling the maximum distance from the agent at which the resources (wood, metal, deer, and food) can appear. We can gradually increase this distance as the learning progresses, as illustrated the figure on the right, which shows the deer appearing progressively further away during training, thus inducing a curriculum. This type of environment shaping can remove the need for reward shaping, and potentially alleviates its shortcomings. We will use this setting to study hypothesis *H2*.

**Dynamic vs. static environments.** We can construct dynamic versions of both of our tasks by varying the probability that the environment changes at any given time, regardless of the agent's actions. This captures the fact that *natural* environments will change on their own, regardless of what the agent does: other animals will move around, weather will

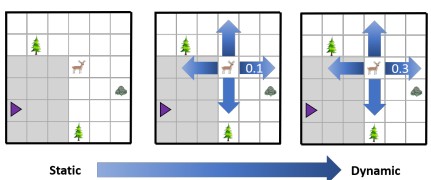

change, etc. In both grid domains, we define a continuous spectrum of dynamic effects, in terms of a *dynamic event probability*. In the tool-making task, the dynamic effect probability $p$ controls the resource generation, determining the probability that a resource will spawn in empty squares no less than two squares away from the agent at each time step. To keep the expected quantity of resources constant throughout the trajectory we allow resources to decay (disappear) after a lifespan of $\frac{I}{p}$ timesteps, where $I$ is the number of instances of this resource that appear in the initial environment. In the hunting task, the dynamic effect probability $p$ is the probability that a deer will move to a random adjacent square on each time step, as illustrated in the figure above. The static version of these environments will have $p = 0$, such that resources are in fixed positions and only respawn as needed when the agent completes the task, and deer do not move. In the Unity Food Collector environment, the dynamic property is given by the speed at which both healthy and poisonous food move, which was tested across the range of $v = 0$ to $v = 16$, corresponding to the velocities set for the food objects in the Unity engine. The static setting here corresponds to stationary food. By studying dynamic and static environments in episodic and non-episodic settings, we can analyze hypothesis *H3*.

## 4.1 EVALUATION

In order to compare agents trained under the different environment conditions, we must construct a single consistent evaluation protocol. We use the same agent network architecture and RL algorithm for all experiments, with details provided in Appendix A. We evaluate all agents on a set of validation tasks that are chosen to be as close as possible to the "standard" RL setting, which is episodic and static. We generate 100 validation tasks by randomly generating environments with varying initial resource and agent locations. Performance is measured by the proportion of validation tasks solved. We first study non-episodic learning and dynamic vs. static environments by varying these training settings with sparse reward and no environment shaping. We will then study reward shaping and environment shaping in the dynamic non-episodic setting. The training settings are:

**Static episodic:** In this setting, $p$ is set to 0 so positions of the resources will be static unless the agent moves them. When the agent finishes the task, the environment is reset to a random initial configuration. The environment is also reset when the episode length reaches 200.

**Static non-episodic:** Here, $p = 0$ and there are no resets. When the agent has used up the available resources, more resources are generated randomly.

**Dynamic non-episodic:** This setting explores the lifelong case with a changing environment, where the dynamic property $p$ is varied between 0 and 1. For tool-making, $p$ is the probability of resources spawning at each timestep in a random location. There is initially two of each type of resource. This environment is lifelong and does not reset when the agent completes the task.

**Dynamic episodic:** This is the same as the dynamic non-episodic version except that the environment is reset to a random initial configuration when the agent completes the task.

## 5 EXPERIMENTAL RESULTS

To study the hypotheses in Section 3, we perform experiments where we train RL agents on the tasks described above and vary different properties of the environment during training.

### 5.1 NON-EPISODIC LEARNING IN DYNAMIC AND STATIC ENVIRONMENTS

In the real world, environments are generally dynamic and non-episodic, meaning that the agent is never reset to an initial state distribution, and many parts of the environment are changing without the agent's intervention. In this section we study the effects of both a dynamic, changing environment and the ability to reset on the learning agent, corresponding to hypotheses *H1* and *H3*. These settings will use sparse reward and no environment shaping.

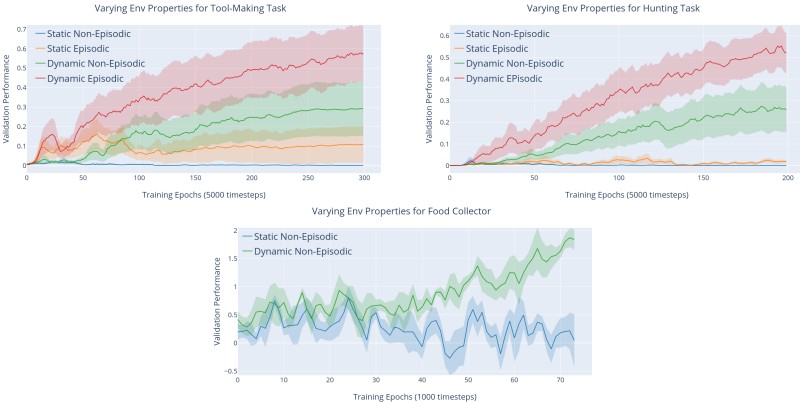

Figure 3: Proportion of validation tasks solved in each setting. Agents learning in static non-episodic environments struggle to learn useful behaviors, while agents learning in dynamic non-episodic environments are substantially more successful. Episodic learning is easier than non-episodic learning on the first task, but non-episodic learning in *dynamic* environments is almost as effective as episodic learning on the hunting task.

**Effect of resets on learning.** In regard to *H1*, we find that learning tasks in static, reset-free, non-episodic environments is difficult. In Figure 3, we compare the performance of the agent trained in each of the four conditions. Recall that all evaluations are conducted in the same setting, with resets and static environments, regardless of how the agent is actually trained. For the tool-making task, we observe that removing resets makes learning more difficult in the typical static case. The agent trained in a static environment without resets obtains the lowest performance ($0\%$ evaluation tasks solved). Adding in resets to the static case helps with performance ($12\%$). In the static environment, we observe that the agent frequently becomes stuck in corners of the map or in areas with no resources.

The results indicate that disabling resets makes the standard static learning condition substantially harder. Indeed, the static no reset agent is unable to learn effectively for either task, even though the static episodic agent does learn the task to a moderate proficiency. However, making the environment dynamic substantially improves performance, in *both* the episodic and non-episodic setting, as shown in Figure 3. These results suggest that dynamic environments to a large extent alleviate the challenges associated with non-episodic learning, confirming hypothesis *H3*. The lesson that we might draw from this is that, although individual properties of natural environments (such as non-episodic learning) can make the learning process harder, combining these properties (i.e., as in the non-episodic dynamic setting) can actually alleviate these challenges, since the dynamics of the environment naturally cause the agent to experience a variety of different situations, even before it has learned to take meaningful and coordinated actions.

**Dynamic environments and non-episodic learning.** We conclude that, when resets are not available, dynamic environments can also help with non-episodic RL. In this section, we study how the frequency of dynamic effects impacts learning. In Figure 3, we can see that making the environment

More details available at `https://sites.google.com/view/ecological-rl`

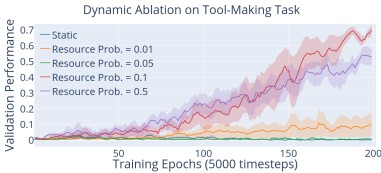 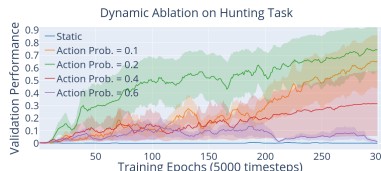

Figure 4: Proportion of validation tasks solved as the dynamic effect probability (i.e., resource probability and deer movement probability) is varied in the non-episodic setting. For all tasks, the standard static environment does not allow for effective learning, but a number of dynamic environment variants allow the agent to learn the task successfully. Evaluation is still carried out in a static environment.

more dynamic increases performance to 70%, compared to the static non-episodic case (0%). Having both a dynamic environment and resets achieves the highest performance (95%), indicating that both are helpful on their own. However, an environment that is too dynamic hinders performance, as we observe in Figure 4, where a dynamic effect probability of 0.5 performs worse than 0.1. This implies that environment dynamics represent a tradeoff: the environment should be stable enough for the agent to learn meaningful behavior, but dynamic enough to present interesting situations. This in some sense resembles the tradeoff typically encountered with exploration constants, e.g. in $\epsilon$-greedy exploration. From this experiment, we can conclude that, although *H3* is generally true, the particular choice of environment settings can greatly impact learning performance. To understand this better, we analyze and compare the effect of "environment shaping," as defined in Section 4, in the following subsection.

## 5.2 Reward Shaping and Environment Shaping

Next, we study how reward shaping and environment shaping compare in terms of their capacity to assist learning in reset-free environments, to study hypothesis *H2*. We perform experiments where we train RL agents on the same tool-making task, in the non-episodic case. We compare shaping the environment during training to shaping the reward function during training. The training environments have one of each resource, spawning at locations sampled uniformly over the world every 20 timesteps which is a much more difficult setting than the ones used in the previous section. We evaluate a range of reward and environment shaping conditions. For all methods the agent is given a reward of 100 each time it completes the task. Resource interaction means picked up a required resource resource. The methods are:

**No shaping with sparse reward:** Resources spawn uniformly and the agent receives task completion reward.

**Distance reward shaping:** The agent is provided with a dense distance-based reward which grants $(-0.01 * \text{distance to nearest required resource})$ and $(1)$ for resource interaction.

**One-time reward shaping:** This is less dense than the distance based reward. The agent is given reward $(1)$ for resource interaction and $-100$ for dropping the resource. This resource reward is only granted the first time and resets every time the task is completed.

**Environment shaping with subgoal reward:** We design a simple shaping method that gradually increases the distance away from the agent at which resources spawn. This distances increases linearly until the resources are placed uniformly over the grid world, as in the shaped reward version. We tried various schedules and found a schedule that starts at a distance of 2 and increases by 1 every $1e5$ environment steps to work well. The subgoal reward is simpler than the one-time reward. The agent is given reward $(1)$ for resource interaction. This reward does not keep track of previous object interactions and grants the bonus multiple times.

**Environment shaping with sparse reward:** We shape the environment as in the previous method but only use the task reward.

**Environment shaping can replace reward shaping.** We compare performance of these methods in Figure 5 for both the episodic and non-episodic case. We find that, even in the episodic case, environment shaping works well and outperforms reward shaping in the long run. Improper reward shaping can alter the optimal policy, thereby biasing learning and resulting in a solution that is worse

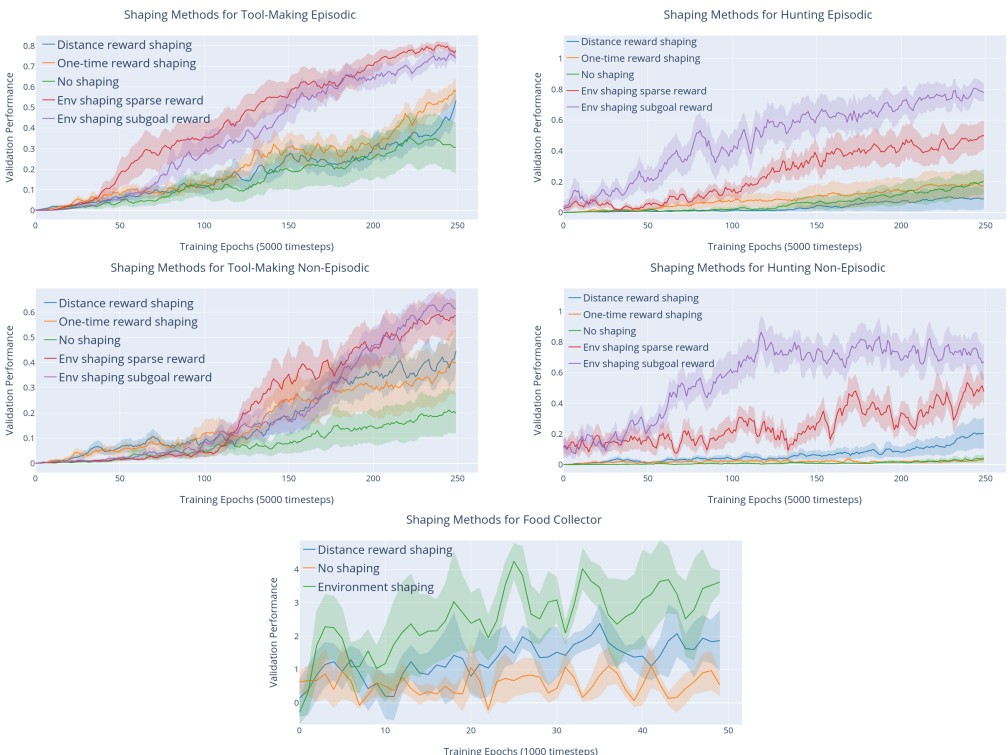

Figure 5: Proportion of validation tasks solved for environment shaping with sparse reward, subgoal reward, and different forms of reward shaping. We see that environment shaping can obtain better final performance than reward shaping.

with respect to the desired performance measure, which typically corresponds to the sparse reward. These behaviors are shown in Appendix D.

Interestingly, we find that environment shaping works better for the more difficult task of *Hunting*. As task complexity grows, so does the difficulty of constructing an unbiased shaped reward for the tasks. In this case, environment shaping benefits from its ease of use and general applicability to various tasks. Further experiments on a harder environment are detailed in Appendix C.

**Human guided environment shaping.** In the real world, environment shaping can be done by doing what we *already* do for other human learners (e.g., children and pupils in school): arranging the environment to be conducive to learning. We also conducted a study of this setting, by having an actual human user interactively specify how the environment should be altered to facilitate the agent's learning process. In this case, the human user was able to provide an environment shaping schedule that outperformed the one we specified manually. Full results for this experiment are provided in Appendix B. These results suggest that environment shaping is not only effective, but is also readily intuitive for a human user to specify interactively, suggesting that it can be a viable way to provide guidance to reinforcement learning agents and may be intuitive to specify, in comparison with reward shaping, which can at times be difficult and counter-intuitive.

## 5.3 EXPERIMENTAL CONCLUSIONS

Our experiments confirm hypothesis *H1* by showing that non-episodic is indeed substantially harder than episodic learning *in standard static environments*. However, our experiments also show that, for all considered tasks, introducing dynamic effects can allow non-episodic learning to succeed, in some cases to a degree that is comparable to the episodic setting, confirming hypothesis *H3*. However, this result is sensitive to the degree of stochasticity, suggesting that the specific dynamics and design of the environment has a large impact on learning. Based on this conclusion, we study how *shaping* the environment influences the learning process, and conclude that appropriate environment

shaping can, in our tasks, supplant the need for more traditional reward shaping, confirming hypothesis *H2*. We further show the human users can effectively select environment shaping schedules manually, suggesting that this is an intuitive way to guide the learning of reinforcement learning agents. Our conclusions support the notion that *ecological* reinforcement learning – the study of the interaction between an RL agent and its environment – is an important topic for further study.

## 6 DISCUSSION

We study how certain properties of natural environments – namely, non-episodic learning without resets, simple "fundamental drive" reward functions, and dynamic environments that evolve on their own even when the agent does not actively intervene – affect the reinforcement learning process. We use the term ecological reinforcement learning to refer to this sort of study, which aims to analyze interactions between RL agents and the environment in which learning occurs. Although these properties by themselves tend to make learning harder, we find that environments that exhibit several of these traits can actually be easier to learn in, and agents trained in such settings can actually outperform agents trained in more conventional episodic settings on the same evaluation tasks. We conclude that in dynamic environments, the variability of situations created by the environment's dynamics and simple rewards that are difficult for the agent to exploit can create a kind of natural curriculum that guides an agent through the emergence of increasingly complex behaviors.

Aside from these potentially surprising observations, the framework of ecological reinforcement learning also points to a new way to approach the design of RL agents. While reward function design is typically considered the primary modality for specifying tasks to RL agents, ecological reinforcement learning suggests that the form and structure of the environment can help to guide the emergence and specification of skills. Combined with the guidance and curricula afforded by natural environments, this suggests that studying and systematizing the interaction between RL agents and various environment properties is an important and interesting direction for future research.

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

## A    AGENT ARCHITECTURE AND TRAINING

We use the same agent network architecture and RL algorithm for all our experiments with minor modification to account for the properties we vary such as reset free RL. Agents are parametrized by an MLP. The environment grid size is $8 \times 8$. The partial grid observation is flattened and processed by a 2 layer MLP of size (64, 64, 32). The inventory observation is processed by a 2 layer MLP of size (16, 16, 16). These outputs are concatenated and then processed by a final MLP of size (16, action_dim). All layers are followed by ReLU nonlinearities except the final layer which uses a softmax to output the action distribution.

We train the agents using double DQN (van Hasselt et al., 2015) and the Adam optimizer (Kingma & Ba, 2014) with a learning rate of 0.0001, selected by sweeping across a range of learning rates, with results shown in Figure 6. Training is done in batch mode such that we alternate between collecting 500 environment steps and taking 500 gradient steps (with batch size 256) over the replay buffer of size 5e5. For environments with resets, the horizon length is set to 200. We swept over various horizon lengths and found 200 to work the best. We also tried setting the horizon length very short (20 and 50) to help with the episodic methods but found no effect. We use epsilon greedy exploration for the policy where epsilon starts at 1 and decays linearly by 0.0001 each timestep to 0.1. For each training method we run 10 random seeds.

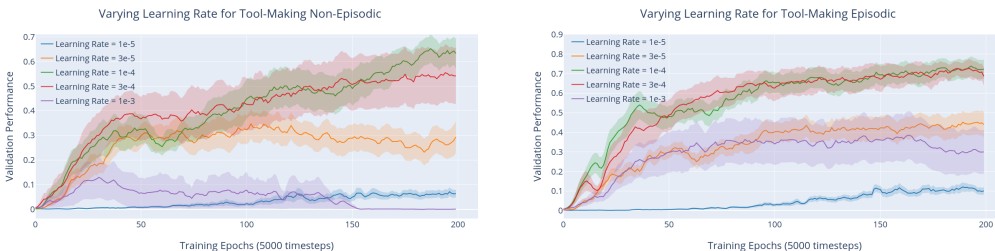

Figure 6: Proportion of validation tasks solved in each setting. Agents learning in static non-episodic environments struggle to learn useful behaviors, while agents learning in dynamic non-episodic environments are substantially more successful. Episodic learning is easier than non-episodic learning on the first task, but non-episodic learning in *dynamic* environments is almost as effective as episodic learning on the hunting task.

## B    HUMAN GUIDED ENVIRONMENT SHAPING

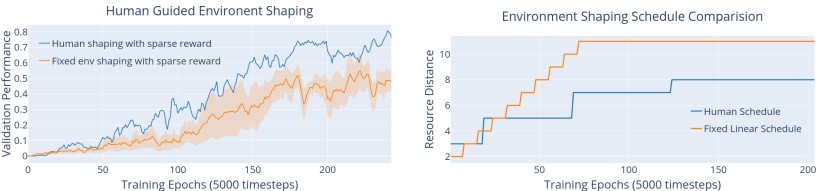

Figure 7: Performance of human guided environment shaping. We ask a human user to interactively shape the environment and observe the human can effectively guide the shaping compared to a predefined environment shaping schedule.

In the real world, environment shaping can be done by humans. In this section, we study if a human user can effectively guide the environment shaping during training, instead of using our predefined curriculum. We use the tool-making task in the non-episodic setting with sparse reward. The form of environment shaping is setting the distance from the agent within which resources can spawn, which can be increased over time to "teach" the agent to reach further-away resources. The human is tasked with providing this distance schedule interactively based on the performance of the agent. At each interaction, the human is given a video demonstrating the agent's current behavior on the training environment and a graph with the agent's validation performance to date. The human user produces two numbers: the resource spawn distance and for how many training epochs to

continue training before requesting another input. This allows the human to adaptively adjust the environment shaping depending on the agent's performance and minimize the amount of human supervision. Interestingly, the human controlled environment shaping does better than our linearly annealed environment shaping, as shown in Figure 7. The human user specifies a slower resource schedule than our programmed environment shaping.

## C  ROBUSTNESS OF ENVIRONMENT SHAPING VS. REWARD SHAPING

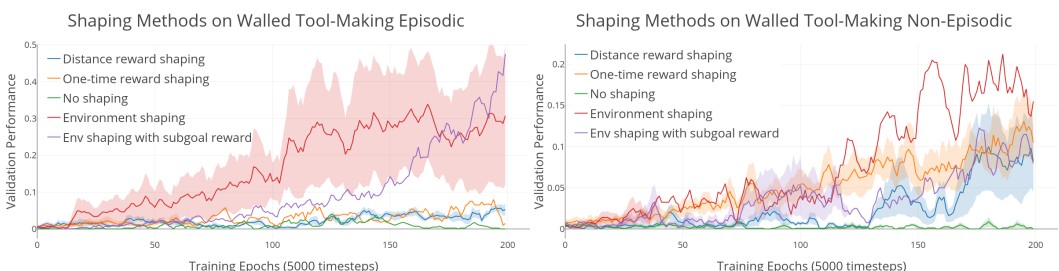

Figure 8: Performance of environment shaping and reward shaping on the axe-making task in an environment with wall obstacles. The distance-based reward suffers while environment shaping, despite operating off of a sparse reward, obtains peak validation performance. We find that this advantage in robustness of environment shaping is present in both the episodic and non-episodic settings, but is enhanced in the former.

We examine the robustness of the different methods of shaping the learning of the agent by studying the performance of the agent in a more challenging environment which contain walls and are more maze-like. This makes the environment less easily navigable and provides more chances for the agent to become trapped in a particular region of the state space. We find in Figure 8 that environment shaping is the best-performing method under this structural challenge under both episodic and non-episodic settings. However, the episodic setting demonstrates a larger gap between the performance of environment shaping and that of reward shaping. We visualize the state visitation counts of the agent under the different shaping methods in Figures 9 (non-episodic) and 10 (episodic) to understand the differences in performance. In the non-episodic setting, the distance-based reward shaping results in the agent getting trapped in corners and therefore spending a high proportion of time there. This demonstrates that reward shaping can be easier to exploit as it alters the true objective. In contrast environment shaping methods results in greater coverage of the grid. This is consistent with their superior performance in Figure 8.

## D  LEARNED BEHAVIOR

In Figure 11, we demonstrate the learned behavior under environment shaping with a sparse reward and reward shaping with the one-time reward. With environment shaping, the agent accomplishes the desired task in 15 timesteps. On the other hand, despite the fact that the one-time reward provides a reward only for the first interaction with the metal, the reward-shaped agent obtains the metal and repeatedly drops and picks it up afterwards, eventually failing to solve it within the allotted 100 timesteps, demonstrating the biasing effect of reward shaping.

In Figure 12, we visualize trajectories from the hunting environment and analyze the learned behavior of two environment-shaped agents, a distance-based reward shaped agent, and a one-time reward shaped agent. The first environment-shaped agent is able to use resources that start out on opposite sides of the world, such that they are never both in view of the agent at the same time. This is notable because the form of environment shaping used is one wherein the agent is provided with resources near it and gradually weaned off over time. The second environment-shaped agent, while presented with a task in which the resources start out on adjacent squares, faces the challenge of the deer moving right before the agent approaches it. We observe that the trained policy is able to make a second attempt at catching the deer, and is successful. The agent trained with distance-based reward shaping displays suboptimal behavior of approaching the axe and then the deer while failing

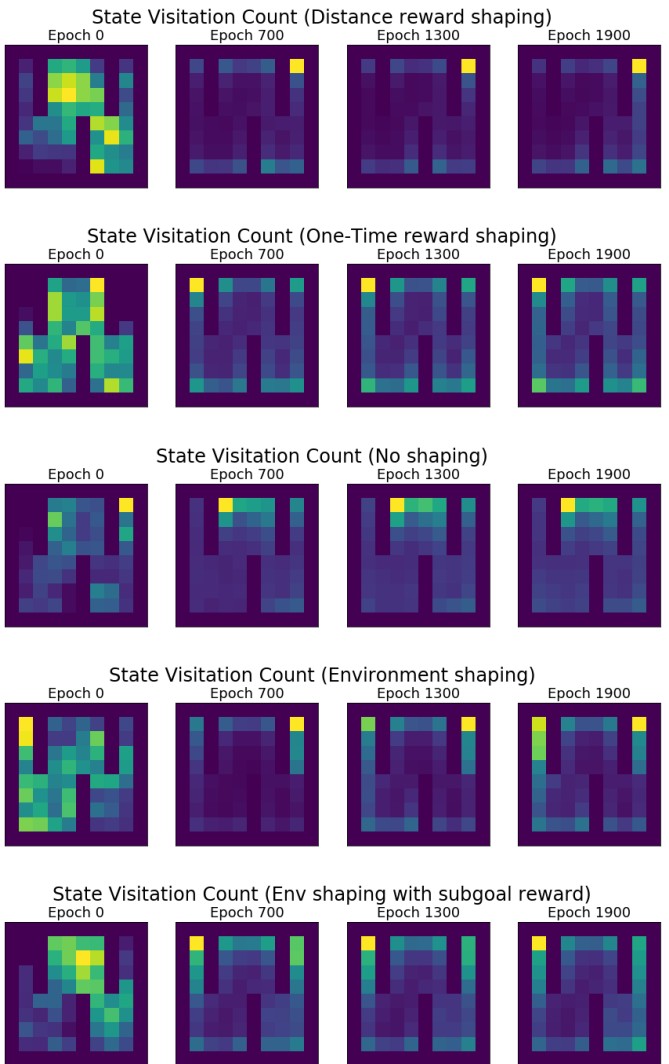

Figure 9: State visitation counts for the non-episodic setting visualized at 4 stages during training under the different methods of shaping. Yellow corresponds to high visitation, dark purple corresponds to low visitation. The darkest purple around the borders and in the map correspond to walls, which cannot be traversed by the agent. We find that the distance-based reward shaping results in the agent getting stuck in the corners of the grid, while the one-time reward and both environment shaping methods result in the most uniform state visitation distribution over the grid during training, indicating that they were able to traverse the grid and explaining their superior performance shown in Figure 8.

to interact with either, which can be seen as a bias resulting from a reward that incentivizes proximity to resources. Finally, the agent trained with one-time reward shaping also shows suboptimal behavior that is explained by the biases of the reward. The agent picks up the axe and then remains stationary throughout the remainder of the trajectory, failing to hunt the deer due to a reward that provides it a small reward bonus for accomplishing the first portion of the task.

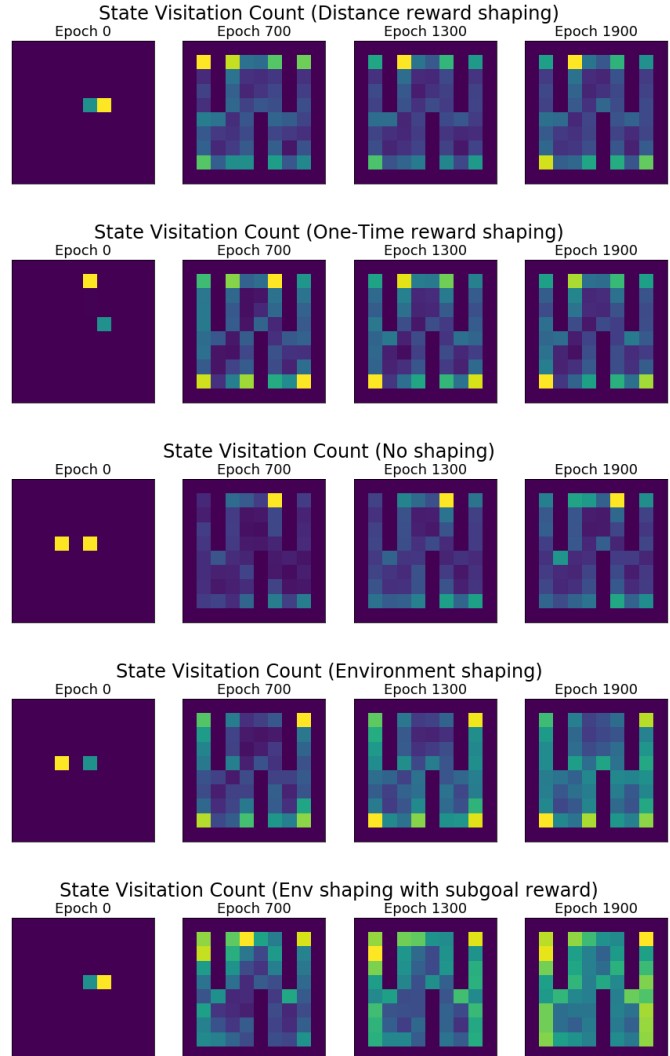

Figure 10: State visitation counts for the episodic setting visualized at 4 stages during training under the different methods of shaping. Yellow corresponds to high visitation, dark purple corresponds to low visitation. The darkest purple around the borders and in the map correspond to walls, which cannot be traversed by the agent. All shaping methods result in a more uniform state visitation distribution than in the non-episodic setting in Figure 9, which aligns with intuition since the resets in the episodic setting help the agent get "unstuck."

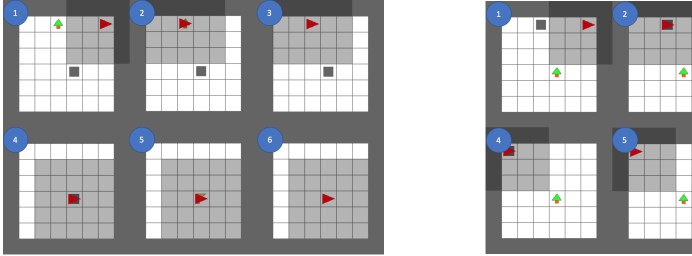

Figure 11: Sample trajectories on validation environments demonstrating learned behavior trained under environment shaping with sparse reward (left) as well as under shaping with the one-time reward (right), both in the non-episodic setting. The shaded region represents the agent's ego-centric partial view of the environment.

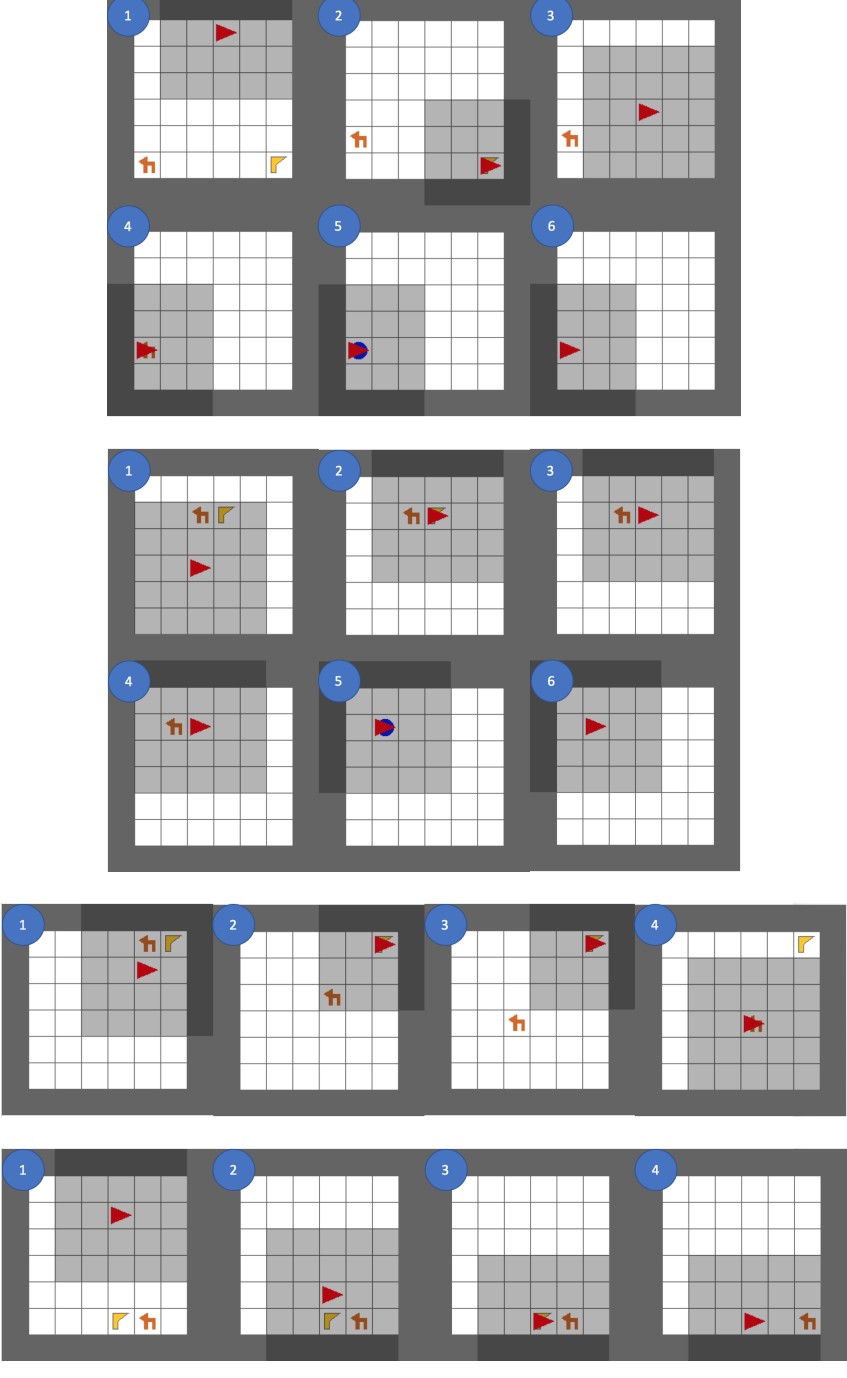

Figure 12: Sample trajectories on validation environments demonstrating learned behavior trained under environment shaping with sparse reward (top two), distance-based reward shaping (third), and one-time reward shaping (bottom), all in the non-episodic setting. While the environment shaped agents accomplish the desired task within 15 timesteps, biased task specification in the last two result in interpretable but suboptimal behavior. The distance-based reward shaped agent goes to the correct resources in order, but without interacting with either. The one-time reward shaped agent picks up the axe, but fails to do anything afterwards. Both reward shaped agents here fail to solve the task within the allotted 100 timesteps.

