# OpenReview forum: "Ecological Reinforcement Learning"
_ICLR.cc/2020/Conference — Reject_

### Official Review · AnonReviewer2 · 2019-10-19
**Official Blind Review #2**

**Rating:** 1

**Review:**

Summary

This paper discusses the value of creating more challenging environments for training reinforcement learning agents. Specifically, the paper focuses on three characteristics of the environment that the paper claims are necessary for developing intelligent agents. The first of these properties is stochasticity in the environment transitions, specifically stochasticity that is independent of the action taken by the agent. The next is sparsity of rewards; discontinuing the use of reward shaping to define desired behavior. Finally the paper argues that environments should not be episodic, that natural environments are continuing tasks so research focus should be around solving continuous tasks.

Review

My primary concern about this paper is the largely insufficient literature review. Many of the claims made in the motivation of this paper are not novel to this paper and are, in fact, incredibly vibrant sub-communities of study within the field of Reinforcement Learning. A more careful literature review should have easily found these communities and more nuanced claims could have been made. I will give concrete examples of the most important cases of missing literature in the following paragraphs, but this list is not exhaustive.

The paper claims that most standard RL environments include detailed reward functions that unnecessarily shape learning and inject bias into the learning process. While I agree that this is problematic, I disagree that this paper provides any novel insights towards this problem. The problem of learning from sparse rewards is well-known in the RL community and is a hot-topic of study. Even in the standard environments cited by the paper we have Montezuma's Revenge and Pitfall, two environment notorious for their difficulty due to sparse reward; each of which with its own host of literature surrounding only the single environment (for instance, I encourage the authors to investigate the highly controversial Go-Explore paper by Uber and its references). Other environments not considered by this paper include the Malmo (Minecraft) learning environment, around which NeurIPS 2019 hosted an extremely sparse reward competition. Another (overlapping) community of RL research interested in the sparse reward setting is the intrinsic reward community, one such paper being Riedmiller and Hafner et al. 2018.

This paper claims that all standard environments are episodic. Of the environments listed as "standard" by this paper, this claim does not even hold. However, there is a large chunk of the RL community that is not represented here. The continual learning and life-long learning communities are focused exclusively on the problem of non-episodic learning. Some example environments used by the community include Malmo, MuJoCo, DeepMind's Lab environment, and many smaller toy domains designed to showcase individual problems including Cart-Pole, RiverSwim, Pendulum, and Acrobot; with the smaller environments from OpenAi Gym cited by this paper. Another smaller community to investigate would be the average reward formulation of the RL problem, which fairly exclusively focuses on the continual learning problem.

Finally, this paper seems vaguely reminiscent of a few particular environments that I have seen in the literature previously. For example, Berkeley's robot task discovery playpen (see for example Singh, Yang, Hartikainen, Finn, Levine 2019). Or an even more similar simulated environment being the Playroom environment by Singh, Barto, Chentanez 2005. Finally, Malmo has been used in a similar way as the tool-building examples mentioned in this paper.

There were a few key issues with the experiments discussed in this paper. The first of which being "Hypothesis 1" which states: "Non-episodic learning is more difficult than episodic learning because the agent must handle a non-stationary learning problem." This hypothesis alone does not appear to be uniformly true. In fact, imagine a simple 5-state random walk markov chain environment without termination. Each state is visited infinitely many times, so the chain is ergodic and there are no non-stationary points. The empirical section uses meta-parameters that were ill-motivated with no discussion about meta-parameter selection. It is critical to point out that the stepsize used for an episodic problem will likely not be the optimal stepsize for the corresponding non-episodic problem, as the magnitude and variance of the considered returns are necessarily different (in response to Figure 3). Further, evaluating over 3 random seeds simply is not sufficient to make any statistically significant claims when comparing any of these curves (for instance in Figure 5).

Additional Comments (do not influence rating)

For a paper exclusively introducing a new control environment, it is critical to include a discussion about the exploration problem. At the very least, I would appreciate seeing sensitivity curves for values of epsilon.

This paper makes many strong claims about the nature of intelligence that are neither supported in the work or are accepted in the community. While it is intuitive that the environment plays a critical role in developing intelligence, the lack of universal definition of intelligence makes this a non-falsifiable claim. Although I appreciate the point the authors are trying to make, which is that RL research frequently is done in the realm of toy simulated domains, I do not think that this paper includes the appropriate supporting evidence to validate such lofty claims.

It would be interesting to change the exploration method from epsilon-greedy to sampling according to the softmax action distribution. This can have dramatically improved performance on non-adversarial exploration problems, and reduces the need for scheduled epsilon decay. It additionally reduces the need for two extra meta-parameters, allowing the empirical claims to be made more strongly without performing some sweep over parameters.

--------------------
Edit after reading discussion, rebuttal, and edits to manuscript.

The paper's intended contribution was different than I had realized during the initial review phase. I appreciate the effort the author put into the response and the changes made to the literature review section of the paper. I think these help to demonstrate the scope and placement of the work considerably.

I still believe the paper over-states the novelty of the results and I still find it difficult to understand their utility. I believe that the entirety of the paper falls under the domain of exploration in RL, but this is not made clear through the introduction or related works section (though the updates to the related works section help considerably).

Stochasticity in the environment helping the agent to learn is not a surprising finding at all if the exploration method is insufficient. To give a small example, imagine an agent wandering in a tabular gridworld. If the agent has no method of exploration, and the environment has sparse reward, then it is not surprising that the agent would get stuck in a corner. If the environment was modified so that each transition had a 5% chance of randomly going another direction (e.g. the "right" action has a 5% chance of becoming an "up" action), then we've effectively encoded epsilon-greedy exploration through the environment dynamics. All of this comes down to say, it is difficult to separate the dynamics of the environment from the agent's exploration method. I think it would require a careful study that considers these aspects more explicitly.

To summarize, I think the paper could easily be accepted to a future conference, but I think it is important to:
- Make connection between the insights and exploration clear, specifically designing the introduction, lit review, and experiments around this connection.
- Make sure the contributions are extremely clear in the writing. Demonstrate those contributions directly in the empirical section.
- Tone down the claims in the writing. Many of the claims about the state of the field in RL are demonstrably incorrect. I agree with the sentiment trying to be expressed, but the absolutism makes it difficult to separate that sentiment from a deep understanding of the current RL literature with only the most important papers cited, or a fundamentally insufficient lit review. To make claims about the state of a field the lit review should be rather extensive.

**Experience Assessment:**

I have published one or two papers in this area.

**Review Assessment: Checking Correctness Of Derivations And Theory:**

N/A

**Review Assessment: Checking Correctness Of Experiments:**

I carefully checked the experiments.

**Review Assessment: Thoroughness In Paper Reading:**

I read the paper at least twice and used my best judgement in assessing the paper.

---

> ### Author Response · Authors · 2019-11-14
> **R2 Response (Part 1)**
>
> Thank you for the detailed and constructive feedback. We have made a number of changes to the paper to address this feedback - including a new environment in Unity, more discussion of related work, experiments with more seeds, and ablations on learning rate. We describe these further in responses to specific comments below:
>
> “My primary concern about this paper is the largely insufficient literature review. Many of the claims made in the motivation of this paper are not novel to this paper and are, in fact, incredibly vibrant sub-communities of study within the field of Reinforcement Learning.”
> > Regarding literature review, we agree we can do more to cite and discuss relevant fields within reinforcement learning. We have included additional references and discussions of differences to work in exploration, continual learning, and lifelong learning in the related work section. In connection to exploration methods, our paper suggests that shaping the environment is an alternative to developing a new algorithm to incentivise exploration in a sparse reward task.
>
> “While I agree that this is problematic, I disagree that this paper provides any novel insights towards this problem. The problem of learning from sparse rewards is well-known in the RL community and is a hot-topic of study.”
> > We agree that learning from sparse reward is well studied within reinforcement learning with many papers proposing exploration methods as solutions including count based methods (Bellemare et al. 2016) and intrinsic motivation (Pathak et al. 2017]. We did not intend to claim that sparse reward environments is not a well studied problem an) that our environments are a new benchmark for sparse reward problems. The findings in our paper indicate that shaping the environment is a simpler alternative to complex exploration methods or reward shaping. We beleive that shaping the environment which in the real world often comes in the form of a parent providing initial resources to their young can help bypass the need for exploration and is easier than devising unbiased shaping the reward schemes. On this point Go-Explore is a relevant paper, however, it still requires building a lower dimensional space with which to search and being able to reset the environment and in their particular case reset to any state while we study the case where resets are not available, even to some initial distribution.
>
> “This paper claims that all standard environments are episodic. Of the environments listed as "standard" by this paper, this claim does not even hold.”
> > Again we emphasize that our contribution is not the introduction of a new environment (https://openreview.net/forum?id=S1xxx64YwH&noteId=SyggDvc7oS) but the findings on an environment which has properties of more natural environments. As far as we know, no paper studies Atari and Mujoco environments in the non-episodic setting. By episodic we mean there exists an absorbing or terminal state where the agent will remain forever and there also exists some mechanism to reset the agent when it reaches the terminal state to some initial state distribution. Atari is naturally episodic because almost all the games have some state of agent death or game over after which the episode resets to the initial state. Even within the continual learning and life-long learning communities, researchers often focus on the case where resets and task boundaries are available (Kaplanis et al. 2018, Rusu et al. 2016, Tessler et al. 2016, Bou-Ammar et al. 2014, Schwarz et al. 2018).

---

> > ### Author Response · Authors · 2019-11-14
> > **R2 Response (Part 2)**
> >
> > "Non-episodic learning is more difficult than episodic learning because the agent must handle a non-stationary learning problem. This hypothesis alone does not appear to be uniformly true.”
> > > We agree that there are cases where non-episodic learning does not create a non-stationary problem. We did not mean to claim that non-episodic learning creates a non-stationary learning problem for all possible MDPs. However there are many cases where non-episodic learning without resets makes the problem more difficult especially in practical settings such as robotics (Eysenbach et al. 2018, Han et al. 2015).
> >
> >
> > “It is critical to point out that the stepsize used for an episodic problem will likely not be the optimal stepsize for the corresponding non-episodic problem”
> > > We have conducted an additional experiment in Appendix Figure 6 where we sweep over stepsizes for both episodic and non-episodic learning. We see that a learning rate of 1e-4 is optimal for both cases (which is the learning rate we use for our other experiments).
> >
> > “Further, evaluating over 3 random seeds simply is not sufficient to make any statistically significant claims when comparing any of these curves (for instance in Figure 5).”
> > > We have rerun our experiments with 10 random seeds and reached the same claims. Figures 3 and Figures 5 have been updated with 10 random seeds.
> >
> > We have also introduced a new environment from the Unity ML-Agents toolkit (Juliani et al. 2018). This is a continuous state space partially observed environment where the agent must maximize the number of healthy food items eaten while avoiding consuming poisonous food. We have updates figures 3, 4, and 5 to reflect these new results. The results confirm our previous claims on the old environments. We have also added videos of the agent on this new environment to the website linked in the paper: https://sites.google.com/view/ecological-rl.
> >
> > Kaplanis, Christos, Murray Shanahan and Claudia Clopath. “Continual Reinforcement Learning with Complex Synapses.” ICML (2018).
> >
> > Bellemare, Marc G., Sriram Srinivasan, Georg Ostrovski, Tom Schaul, David Saxton and Rémi Munos. “Unifying Count-Based Exploration and Intrinsic Motivation.” NIPS (2016).
> >
> > Burda, Yuri, Harrison Edwards, Amos J. Storkey and Oleg Klimov. “Exploration by Random Network Distillation.” ICLR (2018).
> >
> > Pathak, Deepak, Pulkit Agrawal, Alexei A. Efros and Trevor Darrell. “Curiosity-Driven Exploration by Self-Supervised Prediction.” ICML (2017).
> >
> > Rusu, Andrei A., Neil C. Rabinowitz, Guillaume Desjardins, Hubert Soyer, James Kirkpatrick, Koray Kavukcuoglu, Razvan Pascanu and Raia Hadsell. “Progressive Neural Networks.” ArXiv abs/1606.04671 (2016): n. Pag.
> >
> > Tessler, Chen, Shahar Givony, Tom Zahavy, Daniel J. Mankowitz and Shie Mannor. “A Deep Hierarchical Approach to Lifelong Learning in Minecraft.” AAAI (2016).
> >
> > Bou-Ammar, Haitham, Eric Eaton, Paul Ruvolo and Matthew E. Taylor. “Online Multi-Task Learning for Policy Gradient Methods.” ICML (2014).
> >
> > Schwarz, Jonathan, Jelena Luketina, Wojciech Marian Czarnecki, Agnieszka Grabska-Barwinska, Yee Whye Teh, Razvan Pascanu and Raia Hadsell. “Progress & Compress: A scalable framework for continual learning.” ICML (2018)
> >
> > Eysenbach, Benjamin, Shixiang Gu, Julian Ibarz and Sergey Levine. “Leave no Trace: Learning to Reset for Safe and Autonomous Reinforcement Learning.” ICLR (2018).
> >
> > Han, Weiqiao, Sergey Levine and Pieter Abbeel. “Learning compound multi-step controllers under unknown dynamics.” 2015 IEEE/RSJ International Conference on Intelligent Robots and Systems (IROS) (2015): 6435-6442.
> >
> > Juliani, A., Berges, V., Vckay, E., Gao, Y., Henry, H., Mattar, M., Lange, D. (2018). Unity: A General Platform for Intelligent Agents. arXiv preprint arXiv:1809.02627. https://github.com/Unity-Technologies/ml-agents

---

### Official Review · AnonReviewer3 · 2019-10-23
**Official Blind Review #3**

**Rating:** 3

**Review:**

The paper discussed several properties of environments used in reinforcement learning research experiments. The main conclusion is the environment should be dynamic and non-episodic, and the environment shaping is introduced to be effective. In general, the idea is well-presented and easy to follow. However, I have some concerns about the proposed method and experiments:

----
1. In general, I think the arguments in the paper are still a bit vague to be demonstrated using only experimental results. One improvement could be using some mathematical formulation to describe the argument and conducting some analysis. For instance, the paper can formulate the environment shaping and reward shaping concretely and prove that environment shaping could replace reward shaping.


2. For the experiment comparing the reward shaping and environment shaping: the environment shaping method is designed and more complicated than the reward shaping, I think it could be more convincing if the authors investigate and develop more approaches for reward shaping. Otherwise, it is a bit hard to argue the environment shaping could outperform reward shaping a lot.

3. To argue that the non-episodic environment is better than episodic ones, I think the paper should consider more tasks besides the two mentioned in the experiment section. From figure 3, the non-episodic dynamic environment is not very clearly better than episodic one from all scenarios.


**Experience Assessment:**

I do not know much about this area.

**Review Assessment: Checking Correctness Of Derivations And Theory:**

N/A

**Review Assessment: Checking Correctness Of Experiments:**

I assessed the sensibility of the experiments.

**Review Assessment: Thoroughness In Paper Reading:**

I read the paper at least twice and used my best judgement in assessing the paper.

---

> ### Author Response · Authors · 2019-11-14
> **R3 Response**
>
> Thank you for the detailed and constructive feedback. We agree that there could be more mathematical formulation and more tasks. We highlight a connection between mixing times of Markov chains, how dynamic the environment is, and policy gradient learning speed below. We have also added an additional environment in Unity and updated the paper with the new results.
>
> “One improvement could be using some mathematical formulation to describe the argument and conducting some analysis.”
> > There is a connection between dynamic environments and ease of learning. It is known that the learning speed of policy gradient reinforcement learning decreases substantially when the policy explores Markov chains with long mixing times (Morimura et al. 2014, Barlett and Baxter 2000, Baxter and Bartlett 2000; 2001, Kakade 2003). Mixing time quantifies the smallest number of steps needed to approach the stationary distribution from any arbitrary starting state. Properties of the transition matrix determine the mixing time of the Markov chain including conductance (Sinclair 1993). Conductance measures how well connected the state space graph and, “If a chain has significant region in its state space graph that is difficult to enter or leave, then the Markov chain will necessarily take longer to reach the stationary distribution” (March 2011). This is further supported by results from Thodoroff et al. 2018 which suggest that transition matrices with extreme probabilities (low or high) are poorly conditioned, leading to higher mixing times. In our experiments a static environment corresponds to low conductance as the transition matrix has more extreme (zero) probabilities and indeed we see that learning in dynamic environments reaches higher performance compared to static environments (Figure 4). We also see as evidenced by (Thodoroff et al. 2018) that there is a range of intermediate values that is conducive for learning. Learning performance decreases if the dynamic property is increased too high (Figure 4).
>
> “To argue that the non-episodic environment is better than episodic ones, I think the paper should consider more tasks besides the two mentioned in the experiment section. From figure 3, the non-episodic dynamic environment is not very clearly better than episodic one from all scenarios.”
> > We have introduced a new environment from the Unity ML-Agents toolkit (Juliani et al. 2018). This is a continuous state space partially observed environment where the agent must maximize the number of healthy food items eaten while avoiding consuming poisonous food. We have updates figures 3, 4, and 5 to reflect these new results. The results confirm our previous claims on the old environments. We have also added videos of the agent on this new environment to the website linked in the paper: https://sites.google.com/view/ecological-rl. We have also rerun experiments with more seeds (10 seeds) instead of 3. In Figure 3 we now see that dynamic non-episodic clearly outperforms static non-episodic. One of our claims is that dynamic environments make non-episodic learning easier which we do see in the results. We did not mean to claim that non-episodic is always better than episodic learning as having access to resets, generally makes the task easier.
>
> Morimura, Tetsuro, Takayuki Osogami and Tomoyuki Shirai. “Mixing-Time Regularized Policy Gradient.” AAAI (2014).
>
> Bartlett, Peter L. and Jay Baxter. “Infinite-Horizon Policy-Gradient Estimation.” J. Artif. Intell. Res. 15 (2001): 319-350.
>
> Bartlett, Peter L. and Jonathan Baxter. “Estimation and Approximation Bounds for Gradient-Based Reinforcement Learning.” J. Comput. Syst. Sci. 64 (2000): 133-150.
>
> Baxter, Jonathan and Peter L. Bartlett. “Reinforcement Learning in POMDP's via Direct Gradient Ascent.” ICML (2000).
>
> Kakade, Sham M.. “On the sample complexity of reinforcement learning.” (2003).
>
> March, Nathan McNew. “The Eigenvalue Gap and Mixing Time.” (2011).
>
> Thodoroff, Pierre, Audrey Durand, Joelle Pineau and Doina Precup. “Temporal Regularization in Markov Decision Process.” NeurIPS (2018).
>
> Sinclair, Alistair. “Algorithms for Random Generation and Counting: A Markov Chain Approach.” Progress in Theoretical Computer Science (1993).
>
> Juliani, A., Berges, V., Vckay, E., Gao, Y., Henry, H., Mattar, M., Lange, D. (2018). Unity: A General Platform for Intelligent Agents. arXiv preprint arXiv:1809.02627. https://github.com/Unity-Technologies/ml-agents

---

### Official Review · AnonReviewer1 · 2019-10-30
**Official Blind Review #1**

**Rating:** 3

**Review:**

The authors study what they refer to as ecological reinforcement learning, defined as the interaction between properties of the environment and the reinforcement learning agent. They introduce environments with characteristics that reflect natural environments: non-episodic learning, uninformative reward signals, and natural dynamics that cause the environment to change. These factors are shown to significantly affect the learning progress of RL agents and, unexpectedly, the agents can sometimes learn more efficiently in these more challenging conditions.

Clarity:

The paper seems to be clearly written. The code will be made publicly available.

Novelty:

The main contribution from the paper seems to be two novel benchmark problems with characteristics that reflect natural environments and an exhaustive evaluation of the performance of standard algorithms on them. While the experimental results show some light about the performance of existing methods in the proposed environment, the paper does not contain any methodological contributions. Because of this, it is hard to assess the novelty of the work.

Quality and significance:

While the paper makes some interesting points, I feel that the proposed environments are too few, and too simple and unrealistic when compared to real-world problems. Because of this, one cannot know how general and significant the conclusions obtained are.

**Experience Assessment:**

I have published one or two papers in this area.

**Review Assessment: Checking Correctness Of Derivations And Theory:**

I did not assess the derivations or theory.

**Review Assessment: Checking Correctness Of Experiments:**

I assessed the sensibility of the experiments.

**Review Assessment: Thoroughness In Paper Reading:**

I made a quick assessment of this paper.

---

> ### Author Response · Authors · 2019-11-14
> **R1 Response**
>
> Thank you for the constructive feedback. We have updated the paper with a new environment and results. We would also like to point out the many works that use the same environment with which we build off in our comment “Clarification of Contribution” (https://openreview.net/forum?id=S1xxx64YwH&noteId=SyggDvc7oS). These are also listed at https://github.com/maximecb/gym-minigrid.
>
> “ I feel that the proposed environments are too few, and too simple and unrealistic when compared to real-world problems. Because of this, one cannot know how general and significant the conclusions obtained are.”
> > We have introduced a new environment from the Unity ML-Agents toolkit (Juliani et al. 2018). This is a continuous state space partially observed environment where the agent must maximize the number of healthy food items eaten while avoiding consuming poisonous food. We have updates figures 3, 4, and 5 to reflect these new results. The results confirm our previous claims on the old environments. We have also added videos of the agent on this new environment to the website linked in the paper: https://sites.google.com/view/ecological-rl.
>
> Juliani, A., Berges, V., Vckay, E., Gao, Y., Henry, H., Mattar, M., Lange, D. (2018). Unity: A General Platform for Intelligent Agents. arXiv preprint arXiv:1809.02627. https://github.com/Unity-Technologies/ml-agents

---

### Author Response · Authors · 2019-11-09
**Clarification of Contribution**

We thank the reviewers for their detailed comments and feedback. We would like to clarify the contribution of our work. Our contribution is not the introduction of a new environment but rather the empirical findings of varying different environmental properties with standard RL algorithms. While any empirical finding is necessarily specific to a particular domain, showing that adding random dynamics and non-episodic learning together, which both make RL more difficult independently, result in an easier learning problems. This result is in some ways surprising and contradictory to widely held beliefs within the RL community. For example, it is surprising that dynamic environments can make reset free RL easier as most RL algorithms require resets to some initial distribution (Peshkin et al. 2000, Gu et al. 2016) or are necessary in the real world if a robot breaks (Gandhi et al. 2017). We believe these findings could be replicated on existing environments such as Malmo however we chose the particular environment used in the paper for ease of use in modifying dynamics and the capability of building compositional tasks with sparse reward. This form of environment of which we build off has been used in recent papers for language grounding (Chevalier-Boisvert et al. 2019), model based RL (Ke et al. 2019), and exploration in RL (Al-Shedivat et al. 2018). While it is more common for research papers to contain novel algorithms or new theoretical results we would like to point out the value in observational studies about existing methods (Bjorck et al. 2018, Fu et al. 2019) which can give the field greater understanding of important problems and current methods.


Peshkin, Leonid, Kee-Eung Kim, Nicolas Meuleau and Leslie Pack Kaelbling. “Learning to Cooperate via Policy Search.” UAI (2000).

Gandhi, Dhiraj, Lerrel Pinto and Abhinav Gupta. “Learning to fly by crashing.” 2017 IEEE/RSJ International Conference on Intelligent Robots and Systems (IROS) (2017): 3948-3955.

Gu, Shixiang, Ethan Holly, Timothy P. Lillicrap and Sergey Levine. “Deep reinforcement learning for robotic manipulation with asynchronous off-policy updates.” 2017 IEEE International Conference on Robotics and Automation (ICRA) (2016): 3389-3396.

Fu, Justin, Aviral Kumar, Matthew Soh and Sergey Levine. “Diagnosing Bottlenecks in Deep Q-learning Algorithms.” ICML (2019).

Bjorck, Johan, Carla P. Gomes, Bart Selman and Kilian Q. Weinberger. “Understanding Batch Normalization.” NeurIPS (2018).

Chevalier-Boisvert, Maxime, Dzmitry Bahdanau, Salem Lahlou, Lucas Willems, Chitwan Saharia, Thien Huu Nguyen and Yoshua Bengio. “BabyAI: First Steps Towards Grounded Language Learning With a Human In the Loop.” ICLR (2019)

Al-Shedivat, Maruan, Lisa Lee, Ruslan Salakhutdinov and Eric P. Xing. “On the Complexity of Exploration in Goal-Driven Navigation.” ArXiv abs/1811.06889 (2018).

Ke, Nan Rosemary, Amanpreet Singh, Ahmed Touati, Anirudh Goyal, Yoshua Bengio, Devi Parikh and Dhruv Batra. “Learning Dynamics Model in Reinforcement Learning by Incorporating the Long Term Future.” ICLR (2019).

---

### Decision · Program_Chairs · 2019-12-19

**Decision:**

Reject

**Comment:**

This paper investigates how the properties of an environment affect the success of reinforcement learning, and in particular finds that random dynamics and non-episodic learning makes learning easier, even though these factors make learning more difficult when applied individually. The paper was reviewed by three experts who gave Reject, Weak Reject, and Weak Reject recommendations. The main concerns are about missing connections to related work, overstating some contributions, and experimental details. While the author response addressed many of these issues, reviewers felt another round of peer review is really needed before this paper can be accepted; R2's post-rebuttal comments give some specific, constructive, concrete suggestions for preparing a revision.